# Neuroprotective Effects of neuroEPO Using an In Vitro Model of Stroke

**DOI:** 10.3390/bs8020026

**Published:** 2018-02-13

**Authors:** Garzón Fernando, Rodríguez Yamila, García Julio Cesar, Rama Ramón

**Affiliations:** 1Department of Animal Health, University of Nariño, Pasto 52001, Nariño, Colombia; garzongomezfernando@gmail.com; 2Institute of Basic and Preclinical Sciences “Victoria de Girón”, Havana 10400, Cuba; cruz.yamilarod@gmail.com; 3CETEX, CENPALAB, Havana 10800, Cuba; juliocesar.neurotox@gmail.com; 4Department Cellular Biology, Physiology and Immunology, University of Barcelona, 08007 Barcelona, Spain

**Keywords:** stroke, primary cortical neuron culture, excitotoxicity, oxidative stress, neuroEPO, neuroprotection

## Abstract

Erythropoietin (EPO) is a glycoprotein initially identified as a hormone synthesized and secreted by the kidney that regulates erythropoiesis. EPO, and a group of its derivatives, are being evaluated as possible neuroprotective agents in cerebral ischemia. The objective of this study, using an in vitro model, was to determine how neuroEPO—which is a variant of EPO with a low sialic acid content—protects neurons from the toxic action of glutamate. Primary neuronal cultures were obtained from the forebrains of Wistar rat embryos after 17 days of gestation. Excitotoxicity was induced after nine days of in vitro culture by treatment with a medium containing 100 µM glutamate for 15 min. After this time, a new medium containing 100 ng of neuroEPO/mL was added. Morphological cell change was assessed by phase-contrast microscopy. Oxidative stress was analysed by measuring antioxidant and oxidant activity. After 24 h, the treatment with 100 ng of neuroEPO/mL showed a significant (*p* < 0.01) decrease in mortality, compared to cells treated with glutamate alone. neuroEPO treatment decreased mortality and tended to reproduce the morphological characteristics of the control. The oxidative stress induced by glutamate is reduced after neuroEPO treatment. These results confirm that neuroEPO has a protective effect against neuronal damage induced by excitotoxicity, improving antioxidant activity in the neuron, and protecting it from oxidative stress.

## 1. Introduction

Stroke constitutes the third most common cause of death and the most common cause of disability in adults throughout the world [1]. According to the World Health Organization, 15 million people suffer a stroke worldwide each year. Of these, 5 million die and another 5 million are permanently disabled. There are two types of stroke: ischemic strokes, caused by a blockage that cuts off the blood supply to the brain; and haemorrhagic strokes, that occur when a blood vessels burst within or on the surface of the brain. The greater incidence is of ischemic stroke (85%), with a lower mortality than haemorrhagic stroke but with a greater loss of quality of life [2].

The pathophysiology of strokes is complex, and involves mechanisms of excitotoxicity, inflammatory pathways, oxidative damage, ionic imbalances, and apoptosis [3]. Such deleterious states are countered by local and remote protective mechanisms that work to mitigate tissue damage and re-establish homeostasis [4]. Neuroprotection that attenuates or blocks the ischemic cascade and limits neuronal damage has been extensively explored for the treatment of ischemic stroke [5]. In the last two decades, neuroprotective strategies have evolved from targeting a signal neuron pathway to protecting all neurovascular components and improving cell-cell and cell-extracellular matrix interactions that ultimately benefit brain recovery after ischemic stroke [6]. However, despite our increasingly detailed physiological, mechanistic, and imaging characterizations of the ischemic penumbra, no effective neuroprotective therapy has been found so far for the treatment of ischemic stroke. Unfortunately, after many treatments with promising results that appeared potentially capable of recovering from the brain damage caused by ischemic stroke in animal models, none has proved effective in restoring brain function when they are applied in humans suffering ischemic stroke [5]. The current acute neuroprotective approach focusing on the damaging mechanisms at the ischemic penumbra is greatly limited by the complexity and rapid evolution of the cascades of deleterious events involve in the ischemic penumbra [7].

Studies that began more than 20 years ago show that erythropoietin (EPO), a 34 kD glycoprotein that regulates erythropoiesis [8], has neuroprotective effects against ischemic brain injury in both animal experiments [9,10] and clinical studies [11,12,13]. Based on its potent neuroprotection [14] and the fact that it is already widely used in clinical practice, especially in patients with anaemia associated with chronic kidney disease [8], EPO has been considered as a potential candidate for stroke treatment [15]. However, large clinical studies in acute ischemic stroke have failed to demonstrate improved outcome [12,16]. In the earlier clinical study, rhEPO appeared to be beneficial in patients with acute stroke [11]. However, in a subsequent trial, EPO failed to show any beneficial effect in acute ischemic stroke [12]. The use of EPO in the treatment of stroke requires high doses and multiple administrations, which can lead to a high haematocrit and an increase in the number of platelets, which increase the possibility of microcoagulation and secondary infarctions [17]. Therefore, conventional EPO due to the cross-talk with hematopoietic activity may not be suitable for acute stroke treatment.

In the last few years, non-haematopoietic EPO analogues, which include asialoerythropoietin (asialoEPO) [18], carbamylated EPO (CEPO) [19], and rHU-EPO with a low sialic acid content (neuroEPO) [20] that do not exhibit erythropoietic activity, have been tested in the treatment of ischemic stroke.

neuroEPO is a 28 kD recombinant human glycoprotein produced in Chinese hamster ovary (CHO) cells (not a commercial product; the patent is PCT/cu2006/000001 Patent 20050138) supplied by the Centre of Molecular Immunology (CIM, from its Spanish initials) (Havana, Cuba). It is characterized by its low sialic acid content which means that neuroEPO lacks erythropoietic activity while exhibiting neuroprotective properties [20]. neuroEPO can rapidly reach the brain after intranasal delivery [21]. Neuro-EPO has actually proven to be effective in different biomodels of neurodegenerative diseases in preclinical studies. neuroEPO has been shown to exert neuroprotective effects such as improving viability, neurological status and cognitive functions in animal models of stroke [20]. Additionally, this compound is currently used in both transgenic and non-transgenic murine models of Alzheimer´s Disease [22]. To provide further insight into the mechanisms through which neuroEPO could exert its protective effects in cerebral ischemia, we studied the effect of neuroEpo on neuronal death using an in vitro model of stroke.

## 2. Materials and Methods

All experimental procedures were carried out following the guidelines of the Committee for the Care of Research Animals of the University of Barcelona, in accordance with the directive of the Council of the European Community (86/609/EEC) on animal experimentation.

### 2.1. Cell Culture

Cerebral cortex neurons in primary culture were used in this study. Cortical cells were chosen because the cortical area is one of the parts of the brain most frequently affected by stroke. Moreover, the neurons in this area contain a large number of glutamate receptors, so it is prone to excitotoxicity, which plays an important role in stroke [3]. Primary cortical cells were obtained from the cerebral hemisphere of embryos of Wistar rats (Harlan, Spain) after 16–18 days of gestation. The rats were killed with CO_2_ and the foetuses were decapitated. Dissociated cortical cells were plated in cell culture plates at a density of 10^5^ cells/cm^2^ in 9.6 cm^2^ Corning Costar cell culture plates (Madison, WI, USA). Previous to its use, each well was coated with a 1 mL poly-l-ornithine solution (0.01%) for 1 h in the incubator at 37 °C, the poly-l-ornithine solution was removed and washed three times with sterile water. Once the water was removed, the wells were dried in the sterile chamber with ultraviolet light. The cells were cultured in conditioned Neurobasal Medium^®^ (Gibco, Waltham, MA, USA) containing penicillin/streptomycin, enriched with GlutaMAX^®^ (Gibco) and B27^®^ (containing antioxidants), supplemented with 5% horse serum, 5% foetal bovine serum, and 20 mM glucose. Cultures were incubated at 37 °C in a humidified atmosphere containing 5% CO_2_/95% air (pH 7.2). To halt the proliferation of non-neuronal cells, three days after plating, half the culture medium was replaced with medium, as previously described, containing cytosine arabinoside (final concentration: 10 µM). On day 5 after plating, half of the medium was changed with conditioned Neurobasal Medium containing 2 mM glutamine and 2% B27 (AO+), and the neurons were used on day 9.

### 2.2. Glutamate Exposure

On the ninth day in culture the medium was removed, neurons were washed once with pre-warmed (37 °C) with basic saline solution (BSS) containing (in mM) 137 NaCl, 3.5 KCL, 0.4 KH_2_PO_4_, and 0.33 Na_2_HPO_4_·7H_2_O, and treated with conditioned Neurobasal Medium containing 2 mM glutamine and 2% B27 (AO+) containing different concentrations of glutamate (25, 50, 100, and 150 μM) and cells were further incubated for 15 min at 37 °C, in order to establish the concentration most adequate to cause death by excitotoxicity between 40% and 60% of the cells. After this incubation, the medium containing glutamate was removed, and after two washes with BSS, replaced by the Neurobasal medium containing 2 mM glutamine and 2% B27 (AO+), and the cells were further incubated for 24 h at 37 °C.

### 2.3. Cell Death Assays

Neuronal injury was assessed by examination of cultures with phase-contrast microscopy at 20×, while that lactate dehydrogenase (LDH) assay and the MTT 3-(4,5-dimethylthiazol-2-yl)-2,5-diphenyltetrazolium (MTT) assay method were used to assess cell death and cell viability, respectively. Cell death was assessed by measuring the LDH in homogenates of the neurons and in the withdrawn culture medium using the microtiter plate assay described by Dringen [23]. The results were expressed as a percentage of maximum LDH released. Zero percent viability corresponds with 100% of LDH activity in the culture medium. In the MTT assay [24], the formation of a formazan product occurs only in live cells, however, once treated with MMT, the cells are not useable for other measurements. 

### 2.4. Determination of Oxidative Stress

Oxidative stress was analysed by measuring antioxidant and oxidant activity in untreated cortical cells, and in those treated with glutamate alone or with glutamate and neuroEPO. Total antioxidant activity was measured using the Total Antioxidant Capacity Assay Kit (Oxiselect^®^ STA-360) and Total Oxidant Activity Kit via reactive oxygen and nitrogen species (In Vitro ROS/RNS Assay Kit: Oxiselect^®^ STA-347).

### 2.5. Statistics

The normal distribution was analysed using the Shapiro-Wilk test and the homogeneity of the variance by Levene’s test. Data were evaluated by analysis of variance (ANOVA) testing. Differences were considered significant when *p* < 0.05. Results were expressed as the mean ± S.E.M. of *n* = 3 independent experiments. 

## 3. Results

### 3.1. Effect of Different Concentrations of Glutamate on Cell Viability

In order to define the appropriate glutamate concentration to achieve a viability of 40–60%, cortical cells were treated with one of the different concentrations of glutamate (25–150 µM) or no glutamate (control) for 15 min. Once the glutamate was removed, the neurotoxic effect was analysed after 24 h. The results show that increase in glutamate concentrations resulted in a significant decrease in cell viability (Figure 1). Based on these data, all subsequent experiments were carried out at glutamate concentrations of 100 µM for 15 min. 

### 3.2. Effect of Different Concentrations of neuroEPO on Cell Viability

Cortical cells were treated with concentrations of neuroEPO to assess the effect of the neuroEPO on the viability of the cells. The results show a slight decrease in viability in the cell culture treated with neuroEPO compared to untreated cells cultures (control). The differences are not significant (Figure 2).

### 3.3. Effect of Different Concentrations of neuroEPO on Cell Viability after Treatment with Glutamate

To examine the capacity of post-treatment with neuroEPO to protect against glutamate-induced neurotoxicity, primary cortical cells, were exposed to glutamate (100 µM) for 15 min, followed by treatment with different concentrations of neuroEPO (10–100 ng/mL) for 24 h. The results indicate that glutamate exposure induces a significant decrease in cell viability, 40% lower compared to the untreated control cells. This neurotoxic effect caused by the glutamate is significantly reduced by the subsequent treatment with neuroEPO which maintains the cell viability (around 80%) compared to non-treated neurons (control). The protective effect of neuroEPO seems to be similar for the different concentrations of neuroEPO studied here (Figure 3). 

### 3.4. Effect of neuroEPO on Morphological Changes Induced by Glutamate Excitotoxicity in Neuronal Culture

The microscopic observation using phase-contrast microscopy show that, after nine days of incubation, the culture is characterized by many large cells with thick and abundant cellular processes that are connected to neighbouring cells (Figure 4A). Exposure to 100 µM glutamate for 15 min, followed by 24 h in Neurobasal medium shows an evident deterioration of the morphological condition of the culture, characterized by the presence of small, retracted cells with thin or no extensions that are no longer in contact with neighbouring cells (Figure 4B). The treatment with neuroEPO show a tendency to retain the conditions seen in the control: large cells with connections with neighbouring neurons (Figure 4C).

### 3.5. Effect of neuroEPO on Oxidative Stress in Primary Cortical Cells Treated with Glutamate

To detect the presence of oxidative stress, cultured cortical cells were exposed to glutamate (100 µM) for 15 min, followed by treatment with concentrations of neuroEPO (50 ng/mL and 100 ng/mL) for 24 h. Oxidative stress was analysed by measuring the antioxidant and oxidant activity in cortical cells that were untreated, treated only with glutamate, or treated with glutamate and neuroEPO. The results show that there are no significant differences in antioxidant activity between cells treated with glutamate and untreated control cells while, in neurons exposed to glutamate, the treatment with neuroEPO induces a significant increase (*p* > 0.001) in antioxidant activity compared with the oxidant activity observed in untreated control cells and in cells exposed to glutamate and that the antioxidant activity increases as a function of a greater amount of neuroEpo in the medium (Figure 5). 

Cells treated with glutamate show a significant increase in oxidant activity compared with untreated control cells; while the treatment with neuroEPO decreases the oxidant activity induced by glutamate (Figure 5). There was no significant difference between the oxidant activity of the untreated control cells and the cells treated with glutamate plus neuroEPO (Figure 6). 

## 4. Discussion

The major findings of the study are that: (i) neuroEPO can substantially attenuate neuronal death that has been induced by the action of glutamate; and (ii) neuroEPO exposure attenuates oxidative stress induced by glutamate. Thus, neuroEPO favours antioxidant activity and decreases pro-oxidant activity, counteracting oxidative stress triggered by the excessive presence of extracellular glutamate. Our results demonstrate that neuroEPO has the ability to induce antioxidant activity in neurons damaged by excitotoxicity.

The neurotoxic effect of glutamate on the viability of primary cultured neurons has been extensively studied [25,26,27]. In agreement with these studies, our results show that exposure of cultured cortical neurons to 100 µM glutamate for 15 min, followed by the replacement of the medium containing glutamate, by way of control, for 24 h, causes a significant increase in neuronal mortality. Together with high mortality, microscopic observation using phase-contrast microscopy shows that exposure to glutamate causes morphological changes in the neurons [28,29]. We observed the presence of focused swelling of the neuron body with a decrease in the length of the dendrites and the consequent loss of synapses. Treatment, after exposure of the neurons to glutamate, with neuroEPO, mitigates the glutamate-induced neuronal mortality and the morphological changes induced by glutamate. 

In stroke, the lack of oxygen and glucose causes an excessive release of glutamate leading to excitotoxicity. There is abundant evidence that the presence of a state of oxidative stress induced by excitotoxicity plays a pivotal role in cell death in the brain after stroke [30,31]. Oxidative stress is a condition in which cellular antioxidant activity is overwhelmed and is no longer capable of protecting the cell from oxidative damage. This may be due to excessive production of free radicals, loss of antioxidant activity, or both. Our results show that glutamate-induced excitotoxicity leads to an alteration of the redox equilibrium, with neurons increasing their oxidant activity without significant changes in the antioxidant activity, compared with untreated control cells, thus confirming the presence of oxidative stress as already reported by others [32,33]. It is well known that over-activation of NMDA glutamate receptors induced by glutamate toxicity is closely coupled to the generation of nitric oxide (NO) by activation of neuronal nitric oxide synthase (nNOS), an enzyme which is tethered to the NMDA receptor complex by the postsynaptic density protein-95 (PSD95) [34,35], and to calcium overload [36,37]. These cause excessive superoxide production via mitochondrial impairment [37] and NOX (NADPH oxidase) [38,39]. This increased production of free radicals causes the oxidative stress induced by excitotoxicity. 

Treatment with neuroEPO, when applied immediately after a glutamate insult, as in the present study, protects cultured neurons from oxidative stress induced by excitotoxicity. neuroEPO increases cellular antioxidant defences and reduces the oxidant activity in neurons exposed to glutamate. This protective effect seems to be associated with the capacity of neuroEPO to increase antioxidant activity, which allows the neuron to keep its redox balance. Together with averting the loss of the antioxidant capacity of neurons, neuroEPO inhibits the increase of the oxidative capacity induced by glutamate in neurons. Thus, oxidative stress is inhibited by neuroEPO. It has been reported in rat primary cortical neurons that, under conditions of excitotoxicity, EPO reduces the concentration of free intracellular calcium [40]. Excessive accumulation of free cytosolic calcium in neurons, caused by the opening of an excessive number of glutamate receptors in the membrane of the neuron leads to the activation and overstimulation the proteases, lipases, phosphatases, and endonucleases [41,42] followed by the activation of several signalling pathways that cause excessive production of free radicals, mitochondrial damage, and cell membrane disruption, which act synergistically causing apoptotic or necrotic neuron death [36,43]. The ability of the neuroEPO to reduce the accumulation of free calcium in the cytosol may be one of the mechanisms through which neuroEPO carries out its neuroprotective function against excitotoxicity.

In the present study, we show that glutamate can induce morphological changes, oxidative stress, and neuronal death; and that these changes are attenuated by treatment with neuroEPO. Previous studies demonstrated that the neuroprotective effects of EPO require the addition of EPO to the culture several hours prior to glutamate exposure, which suggested that both mRNA and protein synthesis are important [44]. However, our results demonstrate that neuroEPO also has a neuroprotective effect when added after glutamate exposure. This neuroprotective effect is observed 24 h after the addition of neuroEPO to the culture previously exposed to glutamate. From the point of view of possible clinical applications of neuroEPO in the treatment of stroke, this capacity to attenuate neuronal damage when used after stroke represents a more important finding. This is a first in vitro study of the effect of neuroEPO on glutamate excitotoxicity. Our results are similar to those reported previously [18,45,46] using conventional rhEPO. Without ruling out other mechanisms, neuroEPO improves the antioxidant activity of neurons in cultured cortical neurons, thereby protecting against the oxidative stress potentially induced by glutamate. 

It is well known that glycosylation plays a critical role in the biological and pharmacological function of many proteins, affecting their solubility, cellular processing, secretion, and metabolism [19]. NeuroEPO is a protein with low glycosylation. However, our results show that the low degree of glycosylated chain content does not appear to diminish the neuroprotective activity of neuroEPO.

Although results obtained in cell cultures must be extrapolated to animal models with caution, our new findings we report here, coupled with those obtained in animal experiments [20,21], support our hypothesis that neuroEPO contributes to neuronal survival after neuronal damage induced by excitotoxicity.

## Figures and Tables

**Figure 1 behavsci-08-00026-f001:**
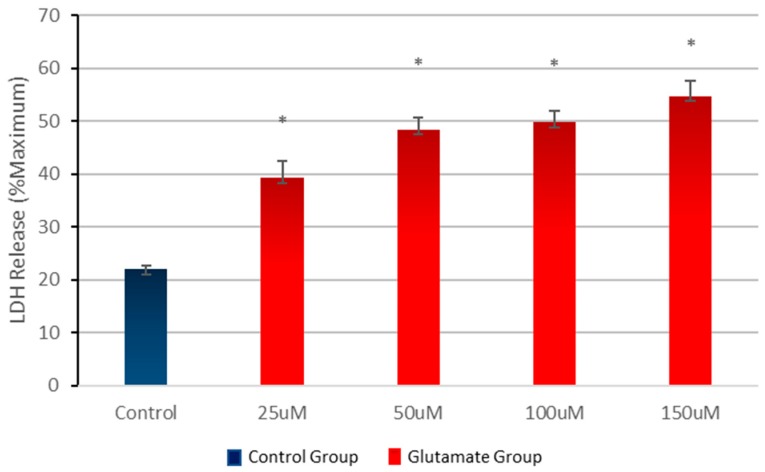
Effects of various concentrations of glutamate on cell mortality. Primary cortical cells were treated with various concentrations of glutamate or in the absence of glutamate (control) for 15 min. Cell mortality was measured after 24 h of treatment. Glutamate causes a significant increase of mortality. Cell mortality is expressed as the mean (±SEM) from three experiments (*n* = 3); * *p* < 0.005 (0–100 mM) compared to untreated control cells.

**Figure 2 behavsci-08-00026-f002:**
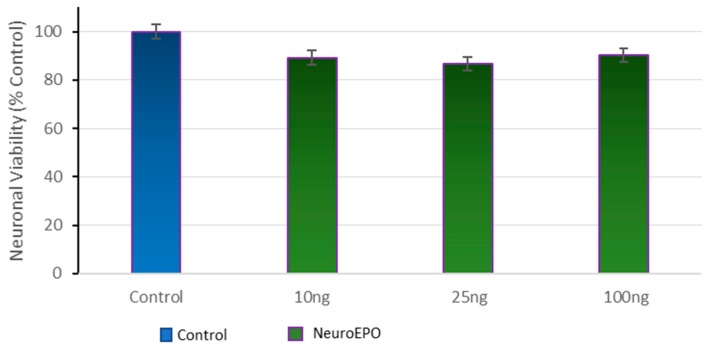
Effects of various concentrations of neuroEPO on cell viability. Primary cortical cells were treated with various concentrations of neuroEPO or in the absence of neuroEPO (control) for 24 h. Cell viability is expressed as the mean (±SEM) from three experiments (*n* = 3).

**Figure 3 behavsci-08-00026-f003:**
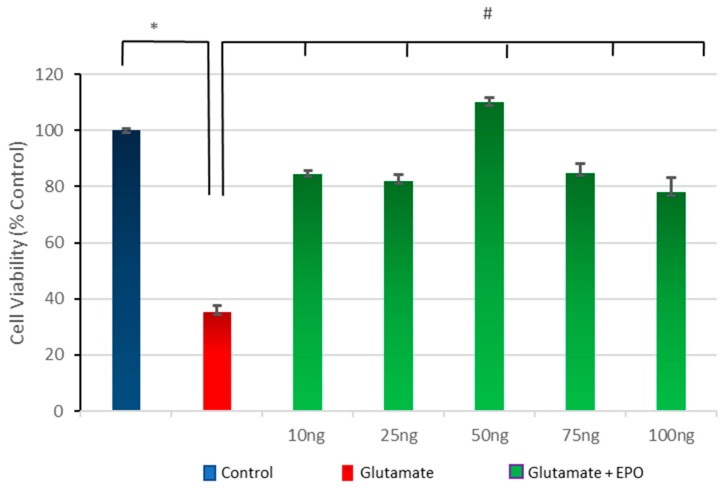
Neuroprotective effect of neuroEPO against glutamate-induced neuronal death. Cortical cells were treated with glutamate for 15 min, followed by treatment with various concentrations of neuroEPO (25, 50, 100, and 150 ng/mL medium) for 24 h. Treatment with glutamate induces a significant (*p* < 0.005) loss of the viability of the neurons compared with the loss of viability of the neurons not treated with glutamate (control). The treatment with neuroEPO significantly (*p* < 0.005) decreases the loss of viability induced by glutamate. The results are expressed as percentages, considering the control as 100%. Data are the mean ± SEM values from *n* = 3 independent experiments; * *p* ≤ 0.005; # *p* ≤ 0.005.

**Figure 4 behavsci-08-00026-f004:**
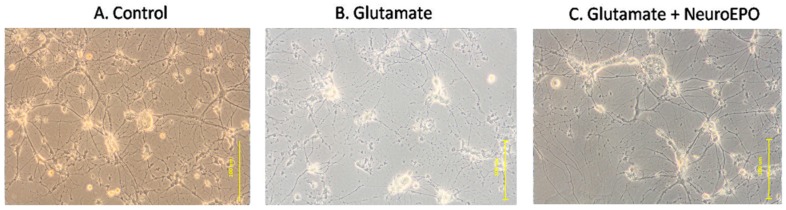
Microscopic observation of primary cortical neurons by phase-contrast microscopy. (**A**) Photomicrographs of cortical neurons after nine days, maintained with Neurobasal medium, show the presence of large cells with thick and thin processes that make contact with the neighbouring cells. (**B**) Photomicrographs of cortical neurons exposed to 100 µM glutamate for 15 min, followed by 24 h in Neurobasal medium. Small cells, retracted with thin processes that do not make contact with neighbouring cells can be observed. (**C**) Photomicrographs of cortical neurons exposed to 100 µM glutamate for 15 min followed by 24 h in Neurobasal medium containing 100 ng/mL of neuroEPO. Cells have thick processes and preserve contact with neighbouring cells. 100× magnification.

**Figure 5 behavsci-08-00026-f005:**
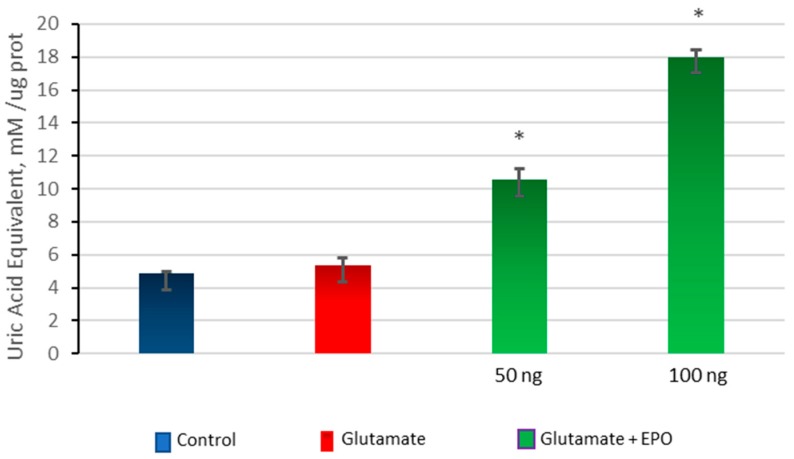
Total antioxidant activity. Cortical cells treated with 100 µM glutamate for 15 min without or with neuroEPO (50 ng/mL or 100 ng/mL) for 24 h. The treatment with glutamate does not significantly modify the levels of antioxidant activity compared with the levels of antioxidant activity of the cells not treated with glutamate (control). NeuroEpo induces a significant increase (*p* < 0.001) in the antioxidant activity, which is dependent on the concentration of neuroEpo, compared both in the cells treated with glutamate and in the untreated cells (control). Data are the mean ± SEM values from 3 independent experiments; * *p* ≤ 0.001.

**Figure 6 behavsci-08-00026-f006:**
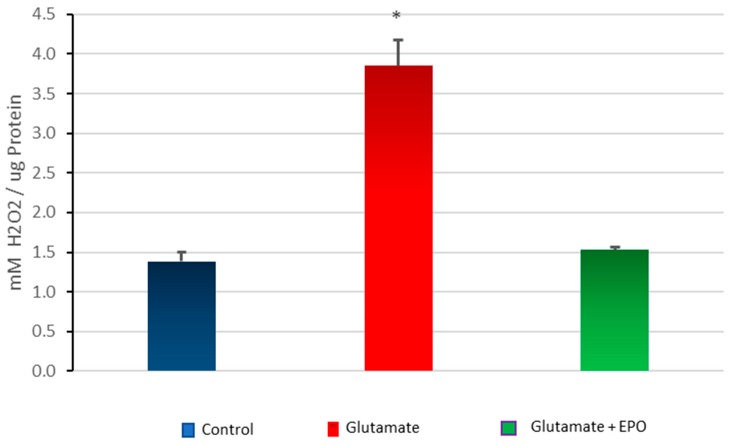
Total oxidant activity. Cortical cells treated with 100 µM glutamate without or with neuroEPO (100 ng/mL) for 24 h. The exposure to glutamate increases the oxidant activity compared with control cells (* *p* < 0.005). The treatment with neuroEpo reduces the oxidant activity caused by glutamate. Data are the means ± SEM values from three independent experiments.

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
