# Peer review of "Neuroprotective Effects of neuroEPO Using an In Vitro Model of Stroke"

_behavsci, 2018, doi:10.3390/bs8020026_

Round 1

Reviewer 1 Report

Overall: Please correct the manuscript for grammatical errors

1)      Introduction

a.       The reference list is very long and extensive – please shorten   

b.      Lines 37, 40, 49: all references are for review papers anyway, only 1 should suffice

c.       Line 57: the statement refers to Epo being widely used in clinical practice – please provide references for human clinical work rather than animal model work

d.      Line 59: same as b)

e.      Lines 69-76 – the authors have not discussed any previous work with neuroEPO at all, please provide a brief and concise rundown of findings for previous studies

2)      Materials and Methods:

a.       Please provide more details about cell culture – how often was the medium changed? How were the cells characterized for purity?

b.      Lines 83-85 need to be the first line of Materials and Methods

c.       Lines 86-87: the line describing stroke should be moved to the results section

d.      Please provide more details about neuronal glutamate exposure  - were cells washed once after removal of glutamate? was conditioned medium used as part of the replacement medium when glutamate exposure ended? Why not?

e.      How were cell density/medium volume discrepancies controlled for during LDH measurements? Was the MTT assay done in parallel for glutamate exposure? As LDH release depends highly on cell density and medium volume, it is important to correct for these factors

f.        Include culture vessel size in Line 101-102

g.       Was a blank measure used for LDH measurement? i.e. As glial medium has serum and serum can have LDH activity, how was this controlled for? 

h.      Statistics – please include a statistics section

3)      Results

a.       Line 112-116, 121-124, 164-166 – A brief sentence describing the experiment would suffice, please remove repeating finer details such as the uM concentrations and cell density etc. All those details need to stay in the results section.

b.    The cultured neurons don't look very convincingly healthy in the images provided (control) - could the authors please provide more images?  

c.      Line 117-120: Please show the dose-response curve of neurons to different glutamate concentrations 

d.       Figure 1: the caption describes cells as being cultured for 11 days whereas the methods & results section both say 9 days. 

e.    Figure 1: the labels on the bars describe ng while the caption and text all refer to uM concentrations. 

f.      Include data about how many replicates/n for each figure/experiment

g.    Please show data from control + Epo for all experiments. This would be helpful in supporting proliferative Vs protective effects of neuroEPO. 

4)      Discussion

a.       It  would be more customary and helpful to start the discussion with a brief description of the overall aim of the study

b.      Line 195: the authors state that ‘neuroEPO can ……….restore the function of synapses’ – the authors have not shown this here in this study so should not be making this claim

c.       Line 197-198 – replace ‘neuronal death’ with ‘oxidative stress’

d.      The authors need to discuss previous findings with neuroEPO specifically and how findings in this study tie in with those.  Was there any available data (either with neuroEPO or EPO) on other cell types with regards to oxidative stress and antioxidant capacity? 

Author Response

Answer to reviewer 2.

1) Introduction

a. The reference list is very long and extensive – please shorten.

The reference list has been shortened by 25% (previously 32 now 24).

b. Lines 37, 40, 49: all references are for review papers anyway, only 1 should suffice.

OK. We have reduced the number of references.

c. Line 57: the statement refers to Epo being widely used in clinical practice – please provide references for human clinical work rather than animal model work.

We provide references for the use of EPO in the clinic.

d. Line 59: same as b).

OK. We have reduced the number of references.

e.Lines 69-76 – the authors have not discussed any previous work with 
neuroEPO at all, please provide a brief and concise rundown of findings for previous studies.

They have been included.

2) Materials and Methods:

a. Please provide more details about cell culture – how often was the medium changed? How were the cells characterized for purity?

Medium was changed two times by week. We have not made a characterization of the purity of the cell culture. Through the use of cytosine arabinonoside, non-neuronal cell proliferation is halted. Other authors using the same methodology (J. Clark Group, Department of Neurochemistry, Institute of Neurology (ION), University College of London) obtained cultures where approximately 97% of the cells were MAP2-positive.

b. Lines 83-85 need to be the first line of Materials and Methods.

OK

c. Lines 86-87: the line describing stroke should be moved to the results section.

OK

d. Please provide more details about neuronal glutamate exposure – were cells washed once after removal of glutamate? was conditioned medium used as part of the replacement medium when glutamate exposure ended? Why not?

After removal of glutamate the cells were washed twice with BSS. The medium used once the glutamate was removed is neurobasal with antibiotics containing 2% B27 and 2 mM glutamine.

When glutamate exposure ended, we do not use conditioned media because contains factors that can influence the effect of the neuroEPO and alter the results in the assessment of the LDH viability, although this difficulty can be overcome by measuring the LDH in that medium.

e. How were cell density/medium volume discrepancies controlled for during LDH measurements? As LDH release depends highly on cell density and medium volume, it is important to correct for these factors.

The method measures the total LDH of the incubation medium and the total LDH of the cellular homogenate. The sum of both is the content of LDH in the well. The quotient between the value of the total LDH content in the medium and the total content of the well x 100 is the % of LDH released to the medium, that is, the mortality in that well. Calculated in this way, the problem pointed out by the reviewer is avoided, but it requires the precise control of the volumes of both the medium and the homogenate of cells. For example, if in well A the cell density is double that in well B, the total LDH content in well A will be double that contained in well B. If the viability is the same, the total of LDH released into the medium in A will be double that in B, but the % of LDH released into the medium will be the same, i.e, the viability is the same in both wells. As a precaution it is recommended to assess the LDH of the medium “No-Cell Control:” Set up triplicate wells without cells to serve as the negative control to determine culture medium background.

Was the MTT assay done in parallel for glutamate exposure?

The analysis of neuronal damage by MMT assay was performed in cells treated with glutamate and neuroEPO after 24 h, in untreated cells (control), and in cells treated only with glutamate after 24 to eliminate glutamate.

f. Include culture vessel size in Line 101-102.  

All cultures were realized using Corning Costar cell culture plate, 6 well, (well bottom area 9,5 cm2). This information has already been included in a previous text, line 58.

g. Was a blank measure used for LDH measurement? i.e. As glial medium has 
serum and serum can have LDH activity, how was this controlled for? .

Yes, is the “no-cell control” cited in e, although the medium used once the glutamate is eliminated, it is Neurobasal with B27 and glutamine, which does not contain serum and has not previously had contact with cells.

h. Statistics – please include a statistics section.

OK. It has already been included.

3) Results

a. Line 112-116, 121-124, 164-166 – A brief sentence describing the experiment would suffice, please remove repeating finer details such as the uM concentrations and cell density etc. All those details need to stay in the results section.

OK

b. The cultured neurons don't look very convincingly healthy in the images provided (control) - could the authors please provide more images?. The images have been replaced by others that we understand are of better quality.

c. Line 117-120: Please show the dose-response curve of neurons to different glutamate concentrations. We have included a new figure (Fig. 1) with the dose-response curve of neurons to different glutamate concentrations.

d. Figure 1: the caption describes cells as being cultured for 11 days whereas the methods & results section both say 9 days.

OK. It´s a mistake. All the experiments have been carried out with 9-day cultures.

e. Figure 1: the labels on the bars describe ng while the caption and text all refer to uM concentrations. All the concentrations of neuroEPO used are in nanograms (ng)

f. Include data about how many replicates/n for each figure/experiment.

The number (n) of replications for each figure is n = 3, which is the number normally used in this class of experiments. In each experiment, three wells were dedicated to the group not treated with glutamate or with neuroEPO (control), three wells to the group treated with glutamate for each concentration, three wells for those treated with neuroEPO and three wells for those treated with glutamate and neuroEPO. Each sample of homogenate from each well and incubation medium are analyzed in triplicate.

g. Please show data from control + Epo for all experiments. This would be helpful in supporting proliferative Vs protective effects of neuroEPO.

A figure with the effect of different concentrations of neuroEpo on cell viability has been incorporated into the manuscript (Fig. 2).

4) Discussion

a. It would be more customary and helpful to start the discussion with a brief description of the overall aim of the study. This opinion of the reviewer is not shared by many other authors. It is also frequent and recommended by experts, start the discussion by pointing out the main contributions of the study.

b. Line 195: the authors state that ‘neuroEPO can ..........restore the function of synapses’ – the authors have not shown this here in this study so should not be making this claim.

We agree with what was indicated by the reviewer. The text has been modified. For the type of experiment, first induce excitotoxicity for 15 minutes and immediately treat with neuroEPo and assess the effect 24 h later, the neuroEPO can reduce the excitotoxic effect of glutamate but we can not say that it restores anything.

c. Line 197-198 – replace ‘neuronal death’ with ‘oxidative stress’.

The whole phrase has been modified.

d. The authors need to discuss previous findings with neuroEPO specifically and how findings in this study tie in with those. Was there any available data (either with neuroEPO or EPO) on other cell types with regards to oxidative stress and antioxidant capacity?

The suggestion has been incorporated into the Discussion. 

Reviewer 2 Report

Abstract

Line 21; Omit “So”

Introduction

Lines 54 and 57 : Comment at Start of sentence “Based on its potent neuroprotection..” is at odds with later comment “However, because the use of EPO in the treatment of stroke requires high doses and multiple administrations, ….” .  This needs to be revised. 

If any clinical stroke trials have been undertaken with EPO or any of its derivatives, the results of these trials should be mentioned.

M&Ms

Please provide details of culture dishes/wells used for culturing neurons.  Also details of pre-coasting of culture vessels.

Base on Fig 2, neurons do not appear well dispersed into individual cells, but clumped.  This suggests culture conditions are not ideal and that neurons may not behave as would be expected (see comment blow).  For example, clumping of the neurons could affect how they are maturing and responding to glutamate in vitro.

Were cultures maintained for 9 days or 11 days (see Fig 1) before exposure to glutamate??

Glutamate excitotoxicity model. This model does not appear to cause the expected level of neuronal cell death following exposure to glutamate.  Based on our own experience, exposure of mature neuronal cultures (9 to 11 days in culture) to 100 µM glutamate for 15 minutes would normally cause at least >90-95% neuronal cell death.  Also by 15 minutes a large proportion of cell death is likely to be irreversible.

In addition, in cultures exposed to glutamate (Fig 2B) neurons do not appear to be degenerating.  Typically following exposure to glutamate neurons start to swell and eventually lyse.  They display a typical fried egg appearance caused by cell body swelling and round and condensation of the nucleus. This model does not appear to reproduce the classic signs of excitototoxic neuronal death.

Results

Line 114: omit “Primary cortical cells isolated from 16–18-day-old foetal 114 rat brain were cultured for 9 days in 6-well plates (106 cells/well)” and transfer to materials and methods section.

Line 122 Omit “primary cortical cells isolated from 16–18-day-old foetal rat brain and then cultured for 9 days were exposed”

Line 133:  “concentrations of neuroEPO (0, 25, 50, 100, and 150µM)” these concentrations do not match concentrations in graph Fig 1.

Line 138: “3.2. Effect of neuroEPO on morphological changes induced by glutamate excitotoxicity in neuronal culture.” See comment above regarding morphological changes to neurons following glutamate excitotoxicity.

Line 159- 172: Does Uric acid equivalent/ug Prot really (fig 3)  =  Oxidative stress was analysed by measuring antioxidant and oxidant activity in cortical cells..” (line 166) and “neuroEPO induces a significant (p< 0.001) increase in antioxidant activity (line 170).

Discussion

Line 195: “restore the formation of synapses..” this was not assessed?

Line 211: Acute calcium influx and activation of calcium sensitive proteases would normally be considered as a major cell death mechanism during severe glutamate excitotoxicity as would normally occur in cultured mature neurons expressing glutamate receptors.  This needs to be discussed in relation to oxidative stress and potential reasons why model may not be truly representative of severe excitotoxicity, but a more milder form of excitotoxicity injury.

Other

Too much material and methods repetition in text of MS and Fig legends.  Also results should be mainly results; there is too much additional background information.

Author Response

Comments and Suggestions for Authors

Abstract
Line 21; Omit “So”: OK

Introduction

Lines 54 and 57 : Comment at Start of sentence “Based on its potent neuroprotection..” is at odds with later comment “However, because the use of EPO in the treatment of stroke requires high doses and multiple administrations, ....” . This needs to be revised.

It has been revised

If any clinical stroke trials have been undertaken with EPO or any of its derivatives, the results of these trials should be mentioned. 

The references of those trials and a commentary of the results have been incorporated into the manuscript

M&Ms

Please provide details of culture dishes/wells used for culturing neurons. Also details of pre-coasting of culture vessels.  The requested details have been incorporated into the manuscript.

Cell culture plate, 6 wells (Costar, Corning, NY, USA) was used for culture neurons. Previously to its use, each well was coat with a 1 ml poly-L-ornithine solution (0,01 %) for 1 hr in the incubator a 37 ° C, remove the poly-L-ornithine solution and wash 3 times with sterile water.  Once the water has been removed, the wells are dried in the sterile chamber with ultraviolet light

Were cultures maintained for 9 days or 11 days (see Fig 1) before exposure to glutamate??. 

The cultures were maintained 9 days before using them to carry out the experiments. We have corrected the error that also appeared in other parts of the text.

Base on Fig 2, neurons do not appear well dispersed into individual cells, but clumped. This suggests culture conditions are not ideal and that neurons may not behave as would be expected (see comment blow). For example, clumping of the neurons could affect how they are maturing and responding to glutamate in vitro.

We agree with the observations regarding fig. 2. However, the appearances of our cultures are very similar to those of other studies (Choi D.W et al., J Neurosci 8: 185-196, 1988). We think that the use of cytosine arabinofuranoside in order to obtain a purer culture of neurons may imply that the lack of astrocytes, whose effect on the growth of neurons we try to compensate using conditioned medium of astrocytes, is the reason why the neurons do not appear well dispersed. In subsequent experiments, without using cytosine, a remarkable improvement is observed. (We attach images)  Evidently, in these there are astrocytes. However, as the reality in the brain is the coexistence between neurons and glia, to know the effect of the neuroEPO we think it may be more appropriate to use mixed cultures with neurons and glia.

               Glutamate excitotoxicity model. This model does not appear to cause the expected level of neuronal cell death following exposure to glutamate. Based on our own experience, exposure of mature neuronal cultures (9 to 11 days in culture) to 100 μM glutamate for 15 minutes would normally cause at least >90-95% neuronal cell death. Also by 15 minutes a large proportion of cell death is likely to be irreversible.

In the opinion of the reviewer, this model does not appear to cause the expected level of neuronal death following exposure to glutamate. However there are abundant studies in which this model is used. Among others, (Choi D.W et al., J Neurosci 8: 185-196, 1988; Chueng et al. Neuropharmacol 1998; Vergun et al. J. Physiol 531:147-163,2001: Zhang et al. BMC Neurosciencw, 6: 13- 2005).

Also, the reviewer objects that the use of 100 μM glutamate for 15 minutes would normally cause at least >90-95 % neuronal cell death. The severity of neuronal viability caused by the exposure of neurons to glutamate is directly proportional to the concentration of glutamate and the duration of exposure. Also to the region of the brain from where the neurons have been obtained: cortex, hippocampus, cerebellum, etc. The results of the different experiments show a great variability in the effect of the glutamate concentration and the duration of the exposure. For example, in primary cortical cells, the use of 5 mM glutamate and exposure time of 3, 6, 8 and 24 hit cause a mortality of 20% in the cells treated with glutamate compared to 12% in the control group (not treated with glutamate) after 24 hours of exposure (Zhang and Bhavnani. BMC Neuroscience, 6: 13-36, 2005). The group of Choi, use 500 μM glutamate for 20 min (Choi D.W et al., J Neurosci 8: 185-196, 1988). Others (Vergun et al. J. Physiol 531:147-163, 2001; Kume et al. Brain Res 756:200-204, 1997) use 100 µM glutamate for 10 min or 100 µM glutamate for 15 min (Vaarmann et al. Cell Death & Disease, 4: e455-e460, 2013). In accordance with what the reviewer points out, we also have the experience that in some experiments, the exposure of neurons after 9 days to 100μM glutamate for 15 min causes mortality greater than 90%.

In addition, in cultures exposed to glutamate (Fig 2B) neurons do not appear to be degenerating. Typically following exposure to glutamate neurons start to swell and eventually lyse. They display a typical fried egg appearance caused by cell body swelling and round and condensation of the nucleus. This model does not appear to reproduce the classic signs of excitototoxic neuronal death.

In Discussion, we write that: We observed the presence of focused swelling of the neuron body with a decrease in the length of the dendrites and the consequent loss of synapses. The images were sent to colleagues in the Cell Biology department with experience in neuron cultures and with the only information that they were neuron cultures. The answer is the one we included in the manuscript. Of course it is a visual assessment of the aspect of the cell cultures. We have not done a specific study on the degradation and formation of synapses or the length of the dendrites. Therefore, in the text we have modified the terms used to indicate the observed morphological modifications.

Results

Line 114: omit “Primary cortical cells isolated from 16–18-day-old foetal 114 rat brain were cultured for 9 days in 6-well plates (106 cells/well)” and transfer to materials and methods section. 

OK

Line 122 Omit “primary cortical cells isolated from 16–18-day-old foetal rat brain and then cultured for 9 days were exposed”. 

OK

Line 133: “concentrations of neuroEPO (0, 25, 50, 100, and 150μM)” these concentrations do not match concentrations in graph Fig 1.

It is an error. The correct data is nanogrmas (ng).

Line 138: “3.2. Effect of neuroEPO on morphological changes induced by glutamate excitotoxicity in neuronal culture.” See comment above regarding morphological changes to neurons following glutamate excitotoxicity. We think we have little to add to our response to the previous comment

Line 159- 172: Does Uric acid equivalent/ug Prot really (fig 3) = Oxidative stress was analysed by measuring antioxidant and oxidant activity in cortical cells..” (line 166) and “neuroEPO induces a significant (p< 0.001) increase in antioxidant activity (line 170).

We do not understand well exactly what is the objection. Does Uric acid equivalent/ug Prot really (fig 3). The correct should say Uric acid equivalent mM/µg protein. We have corrected that data.

The activity antioxidant in neurons is mainly due to activity of superoxide dismutase and glutathione system. However, in our study to assess oxidative stress, it seems more appropriate to measure total antioxidant activity and total oxidant activity.

In relation with “neuroEPO induces a significant (p< 0.001) increase in antioxidant activity (line 170), the text has been slightly modified.

Discussion

Line 195: “restore the formation of synapses.” this was not assessed? We agree with the observation noted. In this study the formation of synapses has not been analyzed. From the presented data, the visual analysis of the microphotographs allows to observe that the exposure to glutamate causes the death of a significant number of cells as well as a loss of contact between neighboring neurons.

Line 211. Acute calcium influx and activation of calcium sensitive proteases would normally be considered as a major cell death mechanism during severe glutamate excitotoxicity as would normally occur in cultured mature neurons expressing glutamate receptors. This needs to be discussed in relation to oxidative stress and potential reasons why model may not be truly representative of severe excitotoxicity, but a milder form of excitotoxicity injury.

               The phrase “Acute calcium influx and activation of calcium sensitive proteases would normally be considered as a major cell death mechanism during severe glutamate excitotoxicity as would normally occur in cultured mature neurons expressing glutamate receptors”, does not appear in our manuscript. However, we agree with the reviewer that “acute calcium influx and activation of calcium sensitive proteases would normally be considered as a major cell death mechanism during severe glutamate excitotoxicity as would normally occur in cultured mature neurons expressing glutamate receptors.”

               The most part of the glutamate receptors are NMDA receptors, which allow the entry of calcium into neurons. Without going into detail on the characteristics of NMDA receptors, the excessive activation of glutamate receptors, together with the activation of nNOS, cause a very high accumulation of calcium in the neurons that induces the activation of several signaling pathways that acting synergistically lead to the neuronal dead.

In relation with the paragraph”This needs to be discussed in relation to oxidative stress and potential reasons why model may not be truly representative of severe excitotoxicity, but a more milder form of excitotoxicity injury”, the reviewer return to his opinion already pointed out in Material and Methods about the suitability and validity of the use of neuron cultures for study of excitotoxicity and the protective effect of NeuroEPO. To the answer already given, here it seems appropriate to point out that the results show that:

               1. The examination of cultures with phase-contrast microscopy performed 24 h after exposure to glutamate shows that glutamate causes morphologic changes in the neurons with the presence of focused swelling of the neuron body with a decrease in the length of the dendrites and the loss of synapsis. The morphologic changes were confirmed by the measurement of LDH, released by damage or destroyed cells inn the extracellular fluid 24 h after glutamate exposure. A series of experiments showed that the specific efflux of LDH induced by glutamate exposure was linearly proportional to the number of neurons damaged or destroyed (Koh and Choi. J. Neurosci. Methods, 20:83-90, 1987).  

               2. The results show that 24h after the exposure to glutamate, there is a significant alteration in the redox balance caused by the increase in oxidant activity without significant changes in antioxidant activity, which define a state of oxidative stress. .

               Preliminary results show a decrease in the expression of anti-apoptotic proteins (Bcl-2) and an increase in the expression of pro-apoptotic proteins (Bax), and the increase of cytochrome c levels in the cytoplasm, in neurons exposed to glutamate after 24 h.

               Taken together, in our opinion, these data confirm that the model of excitotoxicity used is representative of severe excitotoxicity.

Other

Too much material and methods repetition in text of MS and Fig legends. Also results should be mainly results; there is too much additional background information.

We agree with the recommendations outlined here and have tried to plan them in the new version. We hope we have succeeded.

Thank you very much for the suggestions that undoubtedly helped us improve the manuscript.

Reviewer 3 Report

The authors examine the neuroprotective effect of NeuroEPO, variant of the EPO with low sialic acid content, using in vitro model of stroke. Although authors show some significant effects of the NeuroEPO used in their experimental model, the study seems to be underpowered, since the values (means) and statistics were calculated using only three repetitions (“independent experiments”). The authors demonstrated that use of the NeuroEPO compound after the glutamate exposure have a neuroprotective effect, what suggests that it might have possible future therapeutic application. The study is interesting, however, the manuscript is not clearly written and some concerns need to be clarified.

Major comments:

1.            Although the authors present the statistical differences, they did not include any information about statistics/statistical tests used by them.

2.            The authors unnecessarily repeat the methods at the beginning of each paragraph with results. Methods should be clearly stated in the method section, while the result section should contain clearly described results.

3.            The figure captions (for figs 1, 3 and 4) should describe the figures/results and not the methods.

4.            In the paragraph 2.2 authors wrote that neuronal injury was assessed by morphology – they should provide there also criteria which they used for such assessment.

5.            Figure 1. suggests that even when the data were drawn using three independent experiments the results for all the analyzed samples in each group were identical (or almost identical) since the SEM was close to zero. Is it true? In laboratory practice, having the same results in all repetitions is a quite rare case. The authors may alternatively present in the text the numerical values (e.g. mean +- SD). Moreover, there is inconsistency how long the cells were cultured (9 days, like in the text, or 11 days like in the figure caption?).

6.            Paragraph 3.2 is not clearly written (e.g. line 142: what does it mean, “the cells are characterized by many large cells (…)”). The authors also wrote there about morphological analysis – even if it is just a qualitative description – they should provide criteria for such analysis (please see point 4.). Moreover, the difference between control and glutamate (shown on fig. 2 A & B) is not so striking (authors wrote about an evident deterioration 144 of the morphological condition of the culture).

7.            Figure 2.: Authors should include scale bars on the picture (what does it mean 100x magnification? It is the magnification of the whole imaging system? Magnification of the used microscope objective? – simple addition of the scale bar will solve such questions).

8.            Paragraph 3.3.: Inconsistency in the results and method section regarding the length of cell cultures (9 days or 11 days?) The same applies to Fig. 3 and 4). First three sentences of this paragraph (lines 160-164) contain the statements, not results (they should go to the discussion).

9.            The authors begin the discussion section stating that one of the major findings of the study was the restoration of the synapse formation by neuroEPO (line 195). It is an overstatement since the authors did not analyze the formation of synapses (observation of the neurites in contact does not necessarily mean the formation of synapses). Maybe the authors should include additional staining methods (e.g. with synapsin I (e.g. like in Brain Res. 2010 Nov 4; 1359: 44–55) or using other synaptic proteins). Likewise, later they mention that they observed “decrease in the length of dendrites and the consequent loss of synapses” (lines 207-208) it is also an overstatement since neither the length of dendrites or synapses were not directly analyzed.

10.         Last conclusion (line 253-254) of the manuscript stating that this therapeutic approach could revolutionize the treatment of neurodegenerative disorders is also an overstatement since there is still lack of data showing usability of neuroEPO in the treatment of neurodegenerative disorders –authors themselves analyzed only the use of this compound after the toxic action of glutamate.

Minor comments:

-              There are present grammatical and stylistic mistakes in the text, which should be corrected.

-              The differences shown on the figures 1,3 and 4 should be clearly marked using uniform style (e.g. using lines like in the fig. 1)

-              Style of the references is not uniform (e.g. some journal names abbreviated with dots (e.g. Curr. Neurol. Neurosci. Rep.) other without (e.g. J Cereb Blood Flow Metab).

Author Response

Coments and Suggestions for

Authors

The authors examine the neuroprotective effect of NeuroEPO, variant of the EPO with low sialic acid content, using in vitro model of stroke. Although authors show some significant effects of the NeuroEPO used in their experimental model, the study seems to be underpowered, since the values (means) and statistics were calculated using only three repetitions (“independent experiments”). The authors demonstrated that use of the NeuroEPO compound after the glutamate exposure have a neuroprotective effect, what suggests that it might have possible future therapeutic application. The study is interesting, however, the manuscript is not clearly written and some concerns need to be clarified.

Major comments:

1. Although the authors present the statistical differences, they did not include any information about statistics/statistical tests used by them.

The required information has been included.

2. The authors unnecessarily repeat the methods at the beginning of each paragraph with results. Methods should be clearly stated in the method section, while the result section should contain clearly described results.

Following the indication we have made the modifications. It is our wish that they coincide with the proposal of the reviewer.

3. The figure captions (for figs 1, 3 and 4) should describe the figures/results and not the methods.

OK

4. In the paragraph 2.2 authors wrote that neuronal injury was assessed by morphology – they should provide there also criteria which they used for such assessment.

The criterion used was the visual examination of the images obtained by phase contrast microscopy in the way already commented. It is an observation that has been described by others in similar studies. See Choi et al. J. Neurosc. 8 (1): 185-196, 1998.

5. Figure 1. suggests that even when the data were drawn using three independent experiments the results for all the analyzed samples in each group were identical (or almost identical) since the SEM was close to zero. Is it true? In laboratory practice, having the same results in all repetitions is a quite rare case. The authors may alternatively present in the text the numerical values (e.g. mean +- SD). Moreover, there is inconsistency how long the cells were cultured (9 days, like in the text, or 11 days like in the figure caption?).

According to fig 1, the observation of the reviewer is true. However, of course it does not reflect reality. It is a problem of the application used to build the figure from the numeric results. SEM is not close to zero. It varies between values of 5 and 10. We have tried to improve the quality of the figure.

As for how long the cells were cultured before their use in the experiments, the response is 9 days. The correction has already been made.

6. Paragraph 3.2 is not clearly written (e.g. line 142: what does it mean, “the cells are characterized by many large cells (...)”). The authors also wrote there about morphological analysis – even if it is just a qualitative description – they should provide criteria for such analysis (please see point 4.). Moreover, the difference between control and glutamate (shown on fig. 2 A & B) is not so striking (authors wrote about an evident deterioration of the morphological condition of the culture).

The answer to the question of morphological assessment has already been addressed in a previous response. It is a visual appreciation with which you can agree more or less.

The real data on the cell damage caused by glutamate is the viability value of figure 1 of the original manuscript, now fig 3., but above all, the data not shown in paragraph 3.1, which suggestions from one of the reviewers we include as fig 1 in the revised manuscript. All glutamate concentrations applied (25, 50, 100, 150 μM) for 15 minutes, after 24 h showed a significant increase in mortality compared to cells not treated with glutamate.

7. Figure 2.: Authors should include scale bars on the picture (what does it mean 100x magnification? It is the magnification of the whole imaging system? Magnification of the used microscope objective? – simple addition of the scale bar will solve such questions).

Scale bar has been incorporated on the pictures.

8. Paragraph 3.3.: Inconsistency in the results and method section regarding the length of cell cultures (9 days or 11 days?) The same applies to Fig. 3 and 4). First three sentences of this paragraph (lines 160-164) contain the statements, not results (they should go to the discussion).

In reference to the question of 9 or 11 days, it has already been answered. Of 11 times, now we have reduced them, that the age of the crop is mentioned, in 9 times it is indicated that it was after 9 days and in 2 times that it was 11 days. We have no problem in accepting that it is an error. Accepted the same, there is really a difference between using neurons in DIV 9 and DIV 11?. Many studies use neurons between DIV 9 and DIV 11.

9. The authors begin the discussion section stating that one of the major findings of the study was the restoration of the synapse formation by neuroEPO (line 195). It is an overstatement since the authors did not analyze the formation of synapses (observation of the neurites in contact does not necessarily mean the formation of synapses). Maybe the authors should include additional staining methods (e.g. with synapsin I (e.g. like in Brain Res. 2010 Nov 4; 1359: 44–55) or using other synaptic proteins). Likewise, later they mention that they observed “decrease in the length of dendrites and the consequent loss of synapses” (lines 207-208) it is also an overstatement since neither the length of dendrites or synapses were not directly analyzed.

According to the observation. We have modified the content of the paragraph.

In fact, the experiments performed with the neuroEPO are not adequate to assess whether neuroEPO has the capacity to restore the synapses. The effect of the neuroEPO added after the treatment is, according to our results, to attenuate the effect of glutamate, preventing the glutamate from causing a greater neuronal damage.

10. Last conclusion (line 253-254) of the manuscript stating that this therapeutic approach could revolutionize the treatment of neurodegenerative disorders is also an overstatement since there is still lack of data showing usability of neuroEPO in the treatment of neurodegenerative disorders –authors themselves analyzed only the use of this compound after the toxic action of glutamate.

It was not our intention to consider the last paragraph as a conclusion. It was more like a hope for a future that we all want to be as immediate as possible. Find solution to neurodegenerative diseases that affect a significant part of the population. As a conclusion it has been eliminated.

Minor comments:

- There are present grammatical and stylistic mistakes in the text, which should be corrected.

The manuscript has been previously corrected by an expert in correction of scientific manuscripts written in English. We are sorry we cannot do anything except send it to another expert. We will follow the advice of the Behav Sci. publishing.

- The differences shown on the figures 1,3 and 4 should be clearly marked using uniform style (e.g. using lines like in the fig. 1). OK.

- Style of the references is not uniform (e.g. some journal names abbreviated with dots (e.g. Curr. Neurol. Neurosci. Rep.) other without (e.g. J Cereb Blood Flow Metab).

For the style of references, we have used the Mendeley application. We will try to correct the errors manually.

Round 2

Reviewer 2 Report

Manuscript has been improved considerably and is now suitable for publication.

Author Response

Our thanks for your suggestions and comments which have helped us greatly to improve the quality of the manuscript. Thank you very much.

Reviewer 3 Report

The manuscript improved after the corrections made by the authors, but there are still some concerns which need to be clarified:

Specific comments:

1.      Authors provided the requested information regarding the statistical test used by them, but the use of ANOVA for the experiment with just three repetitions is disputable. ANOVA requires normal distribution within groups (may be violated) and homogeneity of the variations between the groups (Important) – did the authors check for this? Probably a better choice would be to use of the non-parametric analysis (like e.g. Kruskal-Wallis).

2.      Since the authors did not provide the criteria for morphological analysis they should rather not use phrase “morphological analysis” because they do not perform any analysis (e.g. in the caption of fig. 4 or in the discussion) but rather “morphological assessment” or simply “microscopic observation” because it was just visual inspection.

3.      Regarding the cell culture length, the authors corrected the data, however, in their answer, they miss my point: my suggestion was not about the significance of the 9 or 11 days of culture – It was about the consistency of the manuscript. The results presented in the manuscript should match the methods. The authors just should be consistent and not use different numbers in those sections.

4.      Although the authors stated that “the manuscript has been previously corrected by an expert in correction of scientific manuscripts written in English” there are still present strange and not clear sentences like e.g.: (lines 175-177)

“The morphological analysis using phase-contrast microscopy show that, after nine days of incubation, the cells are characterized by many large cells with thick and abundant cellular processes that are connected to neighbouring cells.” – I am not an expert in English, but the cells cannot be characterized by many large cells…;

Author Response

Responses to the comments and suggestions of the reviewer 3.

The manuscript improved after the corrections made by the authors, but there are still some concerns which need to be clarified:

Specific comments:

1.      Authors provided the requested information regarding the statistical test used by them, but the use of ANOVA for the experiment with just three repetitions is disputable. ANOVA requires normal distribution within groups (may be violated) and homogeneity of the variations between the groups (Important) – did the authors check for this? Probably a better choice would be to use of the non-parametric analysis (like e.g. Kruskal-Wallis).

Answer:

We agree with reviewer. We have introduced important modifications:

 The normal distribution was analysed using the Shapiro-Wilk test and the homogeneity of the variance by the Levene test. Data were evaluated by analysis of variance (ANOVA) testing. Differences were considered significant when p < 0.05.  Results were expressed as the mean ± S.E.M. of n=3 independent experiments.

2.      Since the authors did not provide the criteria for morphological analysis they should rather not use phrase “morphological analysis” because they do not perform any analysis (e.g. in the caption of fig. 4 or in the discussion) but rather “morphological assessment” or simply “microscopic observation” because it was just visual inspection.

Answer:

We agree with this observation. See the second paragraph of the answer to point 4.

3.      Regarding the cell culture length, the authors corrected the data, however, in their answer, they miss my point: my suggestion was not about the significance of the 9 or 11 days of culture – It was about the consistency of the manuscript. The results presented in the manuscript should match the methods. The authors just should be consistent and not use different numbers in those sections.

Answer:

Agree. The excitotoxicity experiments were performed at 9 days of culture. Writing that they have been done after 11 days of culture has been a mistake. In the revised version that error has been corrected.

4.      Although the authors stated that “the manuscript has been previously corrected by an expert in correction of scientific manuscripts written in English” there are still present strange and not clear sentences like e.g.: (lines 175-177)

“The morphological analysis using phase-contrast microscopy show that, after nine days of incubation, the cells are characterized by many large cells with thick and abundant cellular processes that are connected to neighbouring cells.” – I am not an expert in English, but the cells cannot be characterized by many large cells…;

Answer:

We agree with the comment of the reviewer and following his suggestion, we have made changes in the corresponding text:

“The microscopic observation using phase-contrast microscopy show that, after nine days of incubation, the culture is characterized by many large cells with thick and abundant cellular processes that are connected to neighbouring cells.”

We, as the reviewer, are not experts in English either. For that reason the first version of the manuscript had already been corrected by an expert in the correction of scientific manuscripts written in English by a non-expert in English. Given the suggestions of the reviewers, the second version was sent back to a corrector. In this case, following the suggestion and offer of Behavioral Sciences editors, the manuscript has been sent to correct by the “MDPI English editing service”. We attach the corresponding certificate.
